# Cost-effectiveness of a home-based group educational programme on renal replacement therapies: a study protocol

Steef Redeker,[1] Mark Oppe,[2] Martijn Visser,[1] Jan J V Busschbach,[1] Willem Weimar,[3] Emma Massey,[3] Sohal Ismail,[1] for the 'Nierteam aan Huis' consortium

¹Section Medical Psychology and Psychotherapy, Department of Psychiatry, Erasmus MC, Rotterdam, The Netherlands
²EuroQol Research Foundation, Rotterdam, The Netherlands
³Department of Internal Medicine, Erasmus MC, Rotterdam, The Netherlands

**Correspondence to**
Mr. Steef Redeker;
s.redeker@erasmusmc.nl

## ABSTRACT

**Introduction** Living donor kidney transplantation (LDKT) is the optimal treatment for most patients with end-stage renal disease (ESRD). However, there are numerous patients who cannot find a living kidney donor. Randomised controlled trials have shown that home-based education for patients with ESRD and their family/friends leads to four times more LDKTs. This educational intervention is currently being implemented in eight hospitals in the Netherlands. Supervision and quality assessment are being employed to maintain the quality of the intervention. In this study, we aim to: (1) conduct a cost-effectiveness analysis of the educational programme and its quality assurance system; (2) investigate the relationship between the quality of the implementation of the intervention and the outcomes knowledge, communication and LDKT activities; and (3) investigate policy implications.

**Methods and design** Patients with ESRD who do not have a living kidney donor are eligible to receive the home-based educational intervention. This is carried out by allied health transplantation professionals and psychologists across eight hospitals in the Netherlands. The cost-effectiveness analysis will be conducted with a Markov model. Cost data will be obtained from the literature. We will obtain the quality of life data from the patients who participate in the educational programme. Questionnaires on knowledge and communication will be used to measure the outcomes of the programme. Data on LDKT activities will be obtained from medical records up to 24 months after the education. A protocol adherence measure will be assessed by a third party by means of a telephone interview with the patients and the invitees.

**Ethics and dissemination** Ethical approval was obtained through all participating hospitals. Results will be disseminated through peer-reviewed publications and scientific presentations. Results of the cost-effectiveness of the educational programme will also be disseminated to the Dutch National Health Care Institute.

**Trial registration number** NL6529

## Strengths and limitations of this study

► Working with a state-transition model involves a trade-off between feasibility and transparency of the model and the level of details of real life conditions.
► We do not have a control group in the implementation study, we will use the reported effectiveness of a randomised controlled trial conducted previously.
► By making a dynamic state-transition model, we can model the prevalence and incidence of patients with end-stage renal disease, and consequently the capacity for the facility needs which has not been done before.
► We have high-quality data at our disposal.

and cost-effectiveness.[1] However, there is a significant number of patients who cannot find a living kidney donor and many patients first undergo dialysis before transplantation with a living donor kidney. Interventions are needed to improve access to LDKT.

Research has shown that knowledge of renal replacement therapies (RRT) and communication between patients and their social circle play an important role in the access to LDKT.[2] Studies have shown that a home-based interventional programme had positive effects for patients with ESRD.[3–5] In our transplant centre in Rotterdam, the Netherlands, we have conducted two studies on this home-based educational approach: one randomised controlled trial (RCT) and one cross-over study. The RCT among 163 patients on dialysis showed significant increases in knowledge and communication about LDKT among the patients in the experimental arm who received the home-based education compared with the standard care control arm. Furthermore, there were significantly more actual LDKTs in the experimental group compared with the control group (17 vs 4, p=0.003).[3]

## INTRODUCTION

Living donor kidney transplantation (LDKT) is the optimal treatment for most patients with end-stage renal disease (ESRD) in terms of quality-adjusted life years (QALY), survival

The cross-over trial was aimed at patients who had not previously undergone RRT and who were eligible for transplantation. In the first phase, the experimental arm received the home-based education while the control group waited. In the second phase, the control group also received the education. This study also showed that there was a statistically significant increase in knowledge and communication regarding RRT among patients and invitees after receiving the home-based education. Of the 80 participants, 49 underwent RRT during the 2-year follow-up. Of these, 34 underwent a LDKT, of which 22 were pre-emptive.[4]

Given these positive results, a home-based educational programme for patients with ESRD and their social network is currently being implemented in four regions in the Netherlands. Per region, a regional hospital and a university transplant hospital are implementing the programme. The regional hospitals will target patients who are yet to start RRT, while the university hospitals will target both these patients and dialysis patients. The educators organise the intervention in such a way that they will do 'whatever it takes', in line with one of the basic principles of multisystem therapy (MST), to make this event as patient tailored as possible.[6] Supervision and quality assessment are being employed to maintain the quality of the intervention. The first aim of this study is to evaluate the cost-effectiveness of the education to support continued implementation. In this article, we present the study protocol of a cost-effectiveness analysis of the educational programme and its quality assurance system. The patient population, the standard care, the quality of the educations and the setting differ per hospital. Therefore, our second objective is to investigate the relationship between the quality of the implementation of the programme, as measured by protocol adherence, and outcome. Outcome is defined in terms of knowledge, communication and LDKT activities.

### Hypotheses

Previous research has provided convincing evidence that transplantation costs less, gives a better survival and a higher quality of life compared with dialysis.[7–9] We therefore hypothesise that the relatively small incremental costs of the home-based educational programme should be cost-effective. Since it is desirable that in the future all waiting list patients can benefit from the effects of the home-based intervention, this programme should be part of standard care. Hence, a solid basis of the cost-effectiveness of that educational programme is warranted.

The second hypothesis is that higher protocol adherence among healthcare providers will be associated with more positive effects of the educational interventions. These effects include an increase in knowledge of renal disease and the treatment options, an increase in communication with family/friends about RRTs, an increase in living kidney donation activities and an increase in QALYs. If a relationship between protocol adherence and

effects is shown, a quality assurance system should be an inseparable part of the educational programme.

The third hypothesis is that a full implementation of the educational programme leads to policy implications regarding care for patients with ESRD. Full implementation may affect the need for dialysis centres and transplantation facilities. By modelling the prevalence we can estimate the need for allocating the healthcare budget. We aim to present the outcomes of the model in a budget impact analysis (BIA).

Thus, the main aim of this article is to discuss the protocol of the cost-effectiveness study of the home-based educational programme and of the quality assurance programme which are currently being implemented in the Netherlands. Additionally, potential policy implications of our hypotheses are discussed in this article.

## METHODS AND DESIGN
### Participants and procedure

The implementation study is being conducted in the following regions of the Netherlands: Rotterdam, Amsterdam, Nijmegen and Groningen.

The home-based educational programme is currently being implemented in eight hospitals in the Netherlands; four university transplant hospitals and four regional hospitals. Regional hospitals were included to reach those patients who are yet to start RRT and in this setting, this is the target population. In these hospitals, there is a large dialysis unit but no transplants are conducted. For these hospitals, the inclusion criteria are: ≥18 years of age, are eligible for transplantation and primary RRT required within the coming 12 months. In the regional hospitals, allied health professionals carry out the intervention. The four university hospitals incorporate both a dialysis unit and a transplant centre, therefore in this setting both patients who are yet to start RRT and dialysis patients are the target population. Eligible patients for these hospitals are required to be ≥18 years, currently undergoing RRT or required within the coming 12 months and eligible for transplantation. In the university hospitals, allied health professionals are accompanied by psychologists to carry out the intervention. The distinction between the university hospitals and the regional hospitals is in line with the protocols of the aforementioned cross-over study and the RCT. An estimate of the potential candidates for this implementation is about 50 patients per year per university centre and 20 patients per regional hospital. The implementation study will take 2 years.

If patients have not been able to find living donor candidates in their social network, they will be asked whether they and their social network wish to receive home-based education from healthcare educators. The nephrologist explains to the patients that the educators will provide information about renal diseases, the different types of RRT and their impact on quality of life. Furthermore, the educators can help to discuss the possibilities of living donation within the social network of the patient. If the

patient consents to the intervention in consultation with his/her nephrologist, the healthcare educators contact the patient to make an appointment for the first home session. Patients are supported in inviting their family/friends to the second educational session. After completion of the programme, an evaluation consultation is planned with the nephrologist. The number of patients who do not consent to participate in the study is recorded, as well as the reason for non-participation.

### Patient and public involvement

Patients were involved in the design of the educational programme. When designing the intervention we anchored the patient participation in the project approach by relying on the results of focus groups among 50 patients from the intended target group. Their opinion was sought regarding two methods described in the literature of additional information/coaching: (1) an additional telephone consultation by the transplant doctor[10] and (2) home education where family and friends are invited to receive knowledge about RRT. Eighty-eight per cent of the participants favoured the home-based education over the telephone consultations.[11] Additionally, a patient panel and organisations were involved in the development of the educational programme protocol and materials. Patients were not involved in the design of the cost-effectiveness study.

### The intervention: home-based education

The intervention consists of two sessions at the patient's home. The intervention is carried out by allied health transplantation professionals and psychologists. In the first session, the goals of the educational programme are discussed and the home-based educational meeting will be prepared. The educators will make an inventory of individuals in the patient's social network using a sociogram. This helps open the discussion on who to invite for the second session, the home-based educational meeting. The need for an independent translator is also discussed.

In the second session, the education takes place. The educators organise this session in such a way that they will do 'whatever it takes', in line with one of the basic principles of MST, to make this event as patient tailored as possible.[6] The primary goal of this intervention is educational, therefore, it is not necessary that all the invitees are or become potential donors. The intervention is based on the previous RCT and cross-over studies.[3 4] A more detailed description can be found in a published protocol manuscript.[12]

### Measures

*Knowledge and communication* are evaluated through questionnaires among all patients and for at least one relative/friend in attendance at the home-based educational session. The knowledge about renal disease and RRTs is measured through a validated knowledge questionnaire R3K-T. This 21-item knowledge questionnaire has been developed specifically for kidney disease, and has good psychometric properties.[13] Answer categories are multiple choice and the number of correct answers is summed. The three-item communication questionnaire can be answered on a scale from 1 (completely disagree) to 5 (completely agree). An example item is 'I can talk about renal replacement therapies with my loved ones.' Finally, we assess patients' and invitees' attitude towards RRT through a nine-item questionnaire. This questionnaire can also be answered on a scale from 1 (completely disagree) to 5 (completely agree). An item example is 'I am afraid donation will harm the health of the donor.' The administration of questionnaires will take place at two occasions: (1) prior to the education either during an outpatient visit after signing the 'informed consent form' or during the first session; and (2) shortly after the second session.

*Protocol adherence measures*: After every completed home intervention an independent telephone evaluation is conducted with the patient and a relative/friend who attended the education programme, to measure the degree of protocol adherence of the educators. The independence is guaranteed through an independent party, specialised in treatment adherence measurement (www.Praktikon.nl). Protocol adherence in this implementation is defined as the extent to which the different teams carry out the educational programme as described in the protocol. Measurement is done with an adaptation of the 'Treatment Adherence Measures' (TAM) questionnaire.[14] TAM is scored on a 0–1 scale, where 1 stands for complete protocol adherence. The results of the TAM can be used for research purposes and to give the educators feedback on the quality of their interventions during the implementation phase.

### Cost-effectiveness
#### Costs
The latest published research on costs of dialysis and transplantation in the Netherlands dates back from the late 1990s.[7] Currently, research on costs of dialysis and transplantation is in its final phase. We will use this forthcoming data (De Wit, personal communication). This cost data is of high quality, as it is based on the national database of insurance companies from 2014, which consists of records from 99% of all Dutch citizens. Cost calculations will include costs of dialysis modality, dialysis access, transplant procedure, other hospital costs, primary care costs, mental healthcare, medication outside the hospital, medical devices, healthcare abroad, transport and other costs. Healthcare costs for transplantation include preparatory research, transplant operation, guidance, after care, donor expenses, dialysis procedure, other hospital costs, primary care, mental healthcare, medication outside hospital, medical devices, healthcare abroad, transportation and other costs. These include all the healthcare costs associated with RRT. Since it is recommended that cost-effectiveness analysis is conducted from a societal perspective, we also aim to calculate the productivity costs.[15] Productivity costs will be estimated

with the friction cost method, as recommended in the Dutch guidelines for economic evaluation.[16] Besides that the work situation of patients participating in the study is recorded, some research has been done regarding labour participation of patients with ESRD.[17 18] The costs of the home-based intervention and the quality assurance will be estimated on the basis of the current practice in the implementation. Informal care costs will be estimated from the literature.[19] The costs of the intervention are recorded per centre.

### Effects

In health economics the effects of interventions under evaluation are preferably expressed in QALYs.[15] Research on QALYs of different dialysis modalities is widespread,[8 20 21] but instruments and patient background variables vary. Therefore, we are currently in the process of collecting quality of life data from both patients who are yet to start RRT and dialysis patients prior to the intervention, through the 5-level version of EuroQol-5 Dimension (EQ-5D-5L), the quality of life instrument recommended in the Dutch guidelines for health economics.[16] We also collect quality of life data from patients after the intervention. The educators administer the EQ-5D-5L questionnaire to patients at baseline, 6, 12 and 24 months after the intervention by telephone.

### Markov model

To assess the cost-effectiveness of the implementation of the home-based educational programme, we will build a 'Markov simulation model'. This model will assess the costs and effects of ESRD treatments as it simulates the course of treatment and disease of the patients. The model will have a similar structure as a previously published model on this population,[7] with some important updates and improvements. Unlike that Markov model, which used multivariate and univariate sensitivity analyses,[7] our model will include probabilistic sensitivity analysis using Monte Carlo simulation. Consequently, uncertainty of all values is considered simultaneously, and the uncertainty in each parameter is assumed to possess a probability distribution.[22] The model will run 10 000 simulations in Microsoft Excel, V.2010. By using probabilistic sensitivity analyses we follow current guidelines in health economics.[16 23 24] A Markov modelling technique is applicable because the decision problem involves risk that is continuous over time, the timing of events is important and events may happen more than once.[25] Within a Markov simulation, the time horizon of the study is divided into a number of discrete time periods, the so-called Markov cycles. A Markov process is based on the idea that patients are always in a certain disease state and that they can change between disease states once during each cycle. By assigning effects to each disease state and keeping track of the time patients remained in each disease state, long-term effects can be calculated. For this cost-effectiveness analysis the effects and cost per health state do not change because of the intervention. We expect that there will be more

**Table 1** Overview of parameters and sources in the Markov model

| Parameter | Sources |
| --- | --- |
| Costs<br>1. Medical costs<br>2. Intervention costs<br>3. Costs of productivity losses<br>4. Informal care costs | 1. Cost study of RRT in the Netherlands by de Wit *et al* (forthcoming)<br>2. Are recorded in the current implementation study<br>3. Work situation of patients is recorded in the implementation study and use of (Dutch) literature.[16 17]<br>4. Will be estimated from existing literature[18] |
| Effects (QALYs) | EQ-5D-5L is conducted prior to the intervention and 6, 12 and 24 months after the intervention. |
| Transition probabilities<br>1. Between treatment modalities<br>2. Mortality rate | Estimated from the database of Nefrovisie |
| Incidence rates | Estimated from the database of Nefrovisie |
| Effect size of the intervention | Used from previous studies[3 4] |

EQ-5D-5L, 5-level version of EuroQol-5 Dimension; QALY, quality-adjusted life years; RRT, renal replacement therapy.

LDKTs because of the intervention. Therefore, besides the costs of the programme, only the transition probabilities will change because of the intervention and will be the only difference between the baseline and the post implementation situation. Table 1 shows an overview of the parameters used and the source.

### Model description

A simplified graphical representation of the Markov model showing only the treatment categories, rather than all the individual Markov states is represented in figure 1. Patients continuously enter the model (inflow) at the start of the cycle and can start on haemodialysis, peritoneal dialysis or transplantation. From there they can move between these treatment modalities. Diabetics and non-diabetics are modelled separately since the transition probabilities between the treatment modalities differ between these groups. Since the incidence of kidney failure is increasing, we will also model this in the cost-effectiveness analysis. This will be calculated using data from the database of *Nefrovisie*, a large national database with records of patients with ESRD. As stated before, the model will include probabilistic sensitivity analysis using Monte Carlo simulation. This means that mean numbers of patients per year per treatment modality will be modelled, and the uncertainty surrounding those mean numbers of patients (ie, 95% CIs). Transition probabilities and incidence rates will be based on primary data and will include uncertainty. Costs and utilities will be

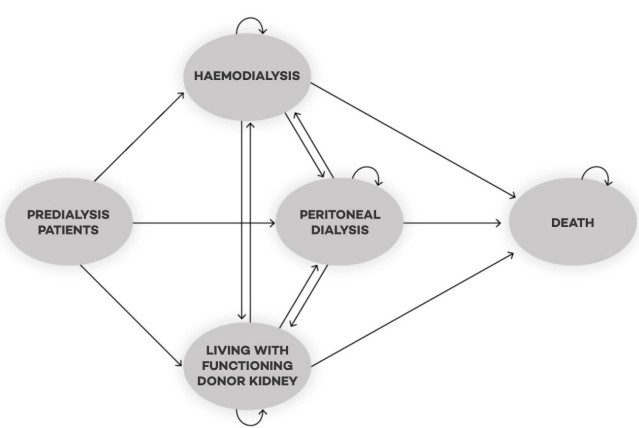

**Figure 1** A simplified graphic representation of the Markov model with the different health states and the transition possibilities between the health states.

included in the model as distributions rather than point estimates.

## Markov states

The Markov states are based on the treatments currently available in the Netherlands. These are: full care centre haemodialysis, limited care centre haemodialysis, home haemodialysis, continuous ambulatory peritoneal dialysis, continuous cyclic peritoneal dialysis, deceased donor kidney transplantation and LDKT. Since transition probabilities and costs may differ over time, that is, a patient who is in his second year of haemodialysis has a different mortality chance than a patient who just started with haemodialysis, we will define separate Markov states for the first year of treatment and subsequent years of treatment for each specific modality. Incident patients who enter the model and prevalent patients who switch between treatment modalities are assigned to the first year Markov states, whereas patients who spend more than 1 year in any health state are transferred to the subsequent years of that same treatment modality.

## Outcomes

The outcome of the implementation will be compared with the baseline situation; the situation before this programme was implemented. The effect size, in terms of an increase in LDKT, will be used from the RCT conducted earlier. Through sensitivity analyses an estimation can be made of the cost-effectiveness of the intervention whether this assumption is either an underestimation or overestimation. A critical assumption will be the extrapolation of the effects after the 2 years.

## Quality assurance

This study has also implemented a central quality assurance system. We hypothesise that the effectiveness is moderated by the protocol adherence of the team of healthcare professionals. In the implementation study, it might be possible that there are differences in the way teams and centres adhere to the protocol. Protocol adherence measures will be assessed by a third party by means

of a telephone interview with the patients and the invitees. Patients and invitees are asked for their opinion and level of satisfaction regarding the way in which the intervention was delivered. Patients and invitees are asked to answer the 15-item TAM scoring list. Items are rated on a Likert scale (1 not at all to 5 very much). Only items that are rated with a 5 will be regarded as fully adherent. Items scored with a 1–4 will be regarded as non-adherent. The outcome of the protocol adherence will be associated with the gain in knowledge and communication skills. We will also look if there is a correlation between the protocol adherence and the amount of LDKTs. A part of the quality assurance is a training that all professionals who conduct the home interventions must take part in. During this training, issues are discussed such as: how to convey uniform and complete information to the patient, how to behave during the home visits, how to create an alignment of the goals of the home visit with the patients, how to assist the patient in inviting friends and relatives, how to deal with emotional moments, how to discuss delicate topics with respect to individual feelings and opinions and, finally, how to ensure no detrimental psychosocial effects of the education occur for the patients and his/her family/friends. All these aspects can be executed in different ways by the educators. A supervisor evaluates the home visits with each team separately every 6 weeks and discuss difficult cases. After these meetings the supervisor is graded by the educators through a 10-item questionnaire regarding the content of the teaching and the interpersonal delivery of the supervisor. Furthermore, the supervisor will bring together all educators for a so-called intervision meeting every 3 months. These intervision sessions are meant to discuss the home visits with each other in order to learn from each other and to keep the procedures similar.

## DISCUSSION

We presented a protocol for assessing the cost-effectiveness of our home-based educational programme and its generalisability. The implementation of the educational programme might both benefit patients and society.

*Cost-effectiveness:* If indeed our hypotheses are confirmed, and the home-based educational programme is cost-effective, then there are convincing arguments to make the programme standard care in the Netherlands. Health insurers already expressed their interest in the programme; this implementation study is supported by *Zorgverzekeraars Nederland* (the Netherlands Health Insurers), which is the 'umbrella organization' of all health insurers in the Netherlands. Additionally, the Dutch Kidney Foundation supports the programme and contributed through three grants in the developmental phases of the home-based educational programme. The Dutch Kidney Foundation is a non-profit organisation which subsidises research and innovation in nephrology and renal transplantation care. Indeed, the health insurance companies have good reason to be interested,

as dialysis is costly. In the Netherlands, 1% of the total healthcare budget is spent on patients with ESRD, who only constitute 0.0006% of the population.[26 27] Furthermore, transplantation is associated with higher quality of life for patients with ESRD compared with dialysis treatment. It is therefore valuable, from both patient and societal perspective, to conduct a complete and extensive cost-effective analysis and consequently to follow-up those results in terms of policy.

### Quality assurance system

Protocol adherence may be of importance to guarantee the effects of the home-based education. First, we expect a positive relation of adherence with outcome in terms of communication, knowledge and the number of transplantations. Second, any problems or regrets of donors and/or patients can only be justified if the evidence-based protocol was followed. The protocol has also been developed after thorough ethical consideration (28), which justified all characteristics of the programme. It can therefore be argued that health insurance makes reimbursement indispensable of the degree of protocol adherence of healthcare suppliers. Moreover, they should facilitate the quality assurance system as an integral part of the programme and ensure that the quality of the interventions is independent of the healthcare suppliers.

### Limitations

Investigating the (cost-)effectiveness of the home-based educational programme has its limitations. In health economics modelling, there is always a trade-off between the feasibility and transparency of the cost-effectiveness model and the level of details of real-life conditions as represented in the model. The more details, the more the model resembles real life, but the downside is that data should be available at that same level of detail and that the model becomes too complex in its feasibility. An example is that we assume that the mortality on dialysis is the same in the second and following years on dialysis. Hence, we know from literature that the mortality chance changes, but the data in later years are scarce and again the model would become more complex, as more 'tunnels states' have to be introduced. We expect that this trade-off will be most prominent in the transition changes between health states. We expect less obvious trade-offs for costs and utility assessment, as we have sufficient data for those variables.

### Cooperation

This investigation is a cooperation between many parties, who all have expressed their support. Obviously, it is possible that this support can be withdrawn for several reasons. For instance, we depend on data from a large national database with records of patients with ESRD (*Nefrovisie*). Hence, much efforts are and will be put in preserving relationships and communication in order to maximise fair successful implementation chances for the programme.

### Ethical considerations

Another challenge that we face is the ethical consideration of promoting living kidney donation through a home-based intervention. Previous studies on the ethics of this argued that such promotion is justified only when the conditions are met, such as (1) participation must be completely voluntary throughout the intervention, (2) no undue pressure should be put on the participants, (3) the education is neutral and non-directive, and (4) the purpose and the procedure should be clear to all participants.[28 29] That does not mean there are no negative consequences whatsoever, but it does mean that the positive outcomes outweigh the negative. It could well be that a patient and/or a donor may regret the decision to have undertaken a transplantation with a kidney of a living donor, and that the donor and/or patient, in hindsight, may have felt undue pressure to donate a kidney. Adherence to the protocol will minimise these potential negative effects. However, it is possible that regret or pressure could lead to negative publicity for the programme. That is a risk since such negative publicity could impede the chances of implementing the programme as standard care.[30] Especially considering that deceased organ donation is currently subject to controversy in the Netherlands since the government and parliament have accepted a donor law. This law is an opt-out system entailing a positive 'no-objection' deceased donor organ donor registration as a default for all Dutch citizens.[31] Given this potential harm due to negative publicity it is crucial to have (1) a protocol which is justifiable from a medical ethical perspective, (2) widespread support from the various organisations involved, and (3) a high quality in terms of protocol adherence and trained adequate educators. If these conditions are met, the quality of the process then justifies its outcome, which is a subtle trade-off between positive and negative outcomes.

### Implementation

Another challenge is the generalisability of the results of the previous effect study on home-based interventions done in Rotterdam. The RCT in the Rotterdam transplant area has shown that the home-based educational programme leads up to four times more LDKTs. The trial took place at the academic transplant centre (Erasmus Medical Centre in Rotterdam) where extensive efforts were already undertaken as part of the standard care to promote LDKTs.[3 32] It is possible that in other transplant areas in the Netherlands, with less experience regarding promoting LDKT, effectiveness in terms of amount of LDKTs may differ. On one hand, this more cautious attitude may lead to lower results than in the Rotterdam transplant area. Hence, organisational conditions within those transplant centres may not optimally facilitate the favourable results of the interventions. On the other hand, if the number of LDKTs was lower than in the Rotterdam area, and the uptake of the intervention is high, the effect could even be higher than in the Rotterdam area. This is

due to a higher effect potential in those centres where living donation was not promoted as actively.

## Learning curve

As with all new programmes, educators will inherently experience a learning curve during the first part of the implementation, which could influence the effectiveness of the programme. For instance, the goal of 50 patients per year per academic transplant centre and the goal of 20 patients per regional hospital may not be reached. Regular supervision (on-the-job) training and peer-to-peer coaching may help overcome this, but a learning curve is unavoidable.

## Policy implications

One of the main pillars of an efficient healthcare system is the ability to provide effective care to patients when needed. It is therefore necessary to have information on the effectiveness of the interventions and their cost to convince policymakers to reimburse the treatment. If the analysis confirms the effectiveness as well as the cost-effectiveness of the home-based educational programme, we recommend that this intervention should be part of standard care.

If the home-based educational interventions would become standard care, this could have several implications. First, it can be expected that patients who are unable to find a living donor will nevertheless profit from an increase number of living kidney donations, as the demand for deceased donor kidneys drops. In other words, the increase in living donation will further lower the waiting list for deceased donor donation as well and thereby increase the chance of a deceased donor donation for those patients without a living donor increases.

Second, the composition of the population of patients with ESRD may change. For instance, it can be expected that the proportion of patients on dialysis will drop and patients with a life-sustaining transplanted kidney will increase. This might have an influence on the demand for dialysis centres and the need for transplantation facilities. We, therefore, aim to incorporate this in a so-called dynamic model to estimate the prevalence over time. When modelling the prevalence, we could make estimates of the need for dialysis centres and transplantation facilities. However, modelling the facility is surrounded by uncertainty. For instance, dialysis centres may have financial incentives to fulfil dialysis capacity. If this would be the case, there will be no monetary benefits for society by increasing the transplant facilities if the proportion of dialysis capacity remains the same. This would mean that there will only be an increase in the average quality of life of patients with ESRD.

Finally, a cost-effectiveness analysis only might not be sufficient to set policy change in motion. Therefore, we anticipate that a policy recommendation accompanied with a BIA will also be required. A BIA addresses the expected changes in the expenditure of a healthcare system after the adoption of a new intervention. It can also be used for budget and resource planning.[33 34]

## CONCLUSION

If our hypotheses are confirmed, we hope by presenting an extensive cost-effectiveness analysis, a BIA, and a policy recommendation that policy change will be set in motion, which again would benefit both patients with ESRD and society.

**Correction notice** This article has been corrected since it first published online. The open access licence type has been amended.

**Acknowledgements** All authors express their gratitude to the participating hospitals, the educators and their supervisors.

**Collaborators** Frederike Bemelman, Karlijn van der Pant, Nicole Zwiers, Marion van Milgen-Adriaens (AMC); Jacqueline van de Wetering, Willem Weimar, Emma Massey, Sohal Ismail, Marian van Noord, Jan van Busschbach, Steef Redeker, Daphne Jansen, Karin Wageveld (EMC); Izaäk Clarisse, Daan Hollander, Petra van de Linde, Cora van Dijk (JBZ); Rene van den Dorpel, Ellen Rovers, Dianne van Dongen, Ton van Kooij, Tahnee van der Marel (Maasstad Ziekenhuis); Harald Brulez, Lobbetje Zwiers, Eline Wisse (OLVG), Ine Dooper, Luuk Hilbrands, Merle Lobeek, Sabine Hopman (Radboud UMC), Jan-Stephan Sanders, Stefan Berger, Jannet Waijer, Merel Kisteman (UMCG); Marco Mahangoe, Margriet Dekker-Janssen, Frieda Hutten, Janet Potgieter (ZGT); Charlotte Boonstra (De Viersprong)

**Contributors** SR drafted the manuscript. MO, MV, JJVB, WW, EM and SI provided significant critical intellectual contributions. All authors reviewed and approved the final version of the manuscript and agree to be accountable for all aspects of the work.

**Funding** The research is funded by the Dutch Kidney Foundation (Nierstichting). The implementation is supported by Dutch Health Insurers (Zorgverzekeraars Nederland).

**Competing interests** None declared.

**Patient consent for publication** Not required.

**Ethics approval** Ethical approval is obtained through all participating hospitals. Results will be disseminated through peer-reviewed publications and scientific presentations. Results of the cost-effectiveness of the educational programme will also be disseminated to the Dutch National Health Care Institute.

**Provenance and peer review** Not commissioned; externally peer reviewed.

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
