## [Reviewer comments · BMJ Open]

ARTICLE DETAILS

TITLE (PROVISIONAL)	Cost-effectiveness of a home-based group educational programme on renal replacement therapies: a Study Protocol
AUTHORS	Redeker, Steef; Oppe, Mark; Visser, Martijn; Busschbach, Jan; Weimar, Willem; Massey, Emma; Ismail, Sohal

VERSION 1 - REVIEW

REVIEWER	Amy D. Waterman, PhD University of California, Los Angeles United States Dr. Amy D. Waterman, PhD owns the intellectual property to the transplant education product Explore Transplant and has licensed it at no-cost to a nonprofit, Health Literacy Media (HLM), who retains all revenue as to their sales. She serves as a consultant to HLM to ensure the accuracy of educational content.
REVIEW RETURNED	27-Aug-2018

GENERAL COMMENTS	The authors are commended for developing, implementing, and evaluating the cost-effectiveness of an educational intervention targeting living kidney donation. However, several major improvements are needed before this protocol paper can be considered suitable for publication. Primary concerns: 1. The structure and organization of the paper causes confusion about what aspects of the methodology apply to this cost-effectiveness study, the implementation study in general, and the development of the education. Is this a protocol paper for the implementation study AND the cost-effectiveness analysis or only the cost-effectiveness analysis? If this paper is intended to serve as a protocol for the implementation study generally, more details are needed as listed here. If not, a concise summary of these items should be included.a. Please provide references or information for what intervention is being implemented as an educational program. What was the educational program based on? Is it based upon the prior RCT and Cross-Over studies mentioned? Is there a prior protocol for this education program implementation?b. "The educators organize the intervention in such a way that they will do 'whatever it takes', in line with one of the basic principles of multisystem therapy (MST)." This statement does not have any references to a prior study nor a reference to a description of what "whatever it takes" principle is in MST.2. The details of the cost-effectiveness analyses provided are not sufficient to justify separate publication.
---

- a. How is the data from the implementation study being used in the cost-effectiveness analysis? What data from the previous RCT will be used?
- b. Why isn't the cost data from the literature included in this protocol? It is part of the input into the cost-effectiveness model.
- c. How will the productivity costs be estimated?
- d. How will the costs of the intervention and quality assurance be estimated?
- e. How many simulations will be run for the Markov model? What are the assumptions that will be made? What are the estimated model parameters and distributions that will be used as input into the model? Which ones are based on the literature and which ones require data from the completed implementation study? These should be listed, perhaps in a table. What sensitivity analyses will be conducted to evaluate robustness of results to incorrect assumptions? What software or programming language will be used to run these models?
- f. The authors state "The outcome of the implementation will be compared with the baseline situation; the situation before this program was implemented." The table of model inputs requested should clearly show what will be the same and will be different in these baseline and post-implementation models.
- g. Here are two examples of cost-effectiveness analyses that provide much more detail on the model specifications:
<https://link.springer.com/article/10.1007%2Fs00198-014-2999-4>
<https://tobaccocontrol.bmj.com/content/23/3/223.long>

Other major comments:

1. The paper is titled "Cost-effectiveness of Educational Interventions Targeting Living Kidney Donation", implying that more than one intervention is being evaluated but this does not seem to be the case. Please clarify.
2. Is the projected sample size sufficient to address to the hypotheses?
3. Quality of life data from patients who are approached for the educational program—Is this using the EQ-5D-5L? When is the questionnaire being administered? The authors state that "we are currently in the process of collecting quality of life data from both pre-dialysis patients and dialysis patients"—is this referring to the implementation study or something separate?
4. "Supervision and quality assessment were employed to map the generalizability of previous research." This meaning of this sentence is not clear.
5. What are the inclusion/exclusion criteria for the implementation study? Particularly for the pre-dialysis patients.
6. The stated second objective is to investigate the relationship between the quality of the implementation of the program and outcome but no analysis methods are described to address this objective.
7. The inclusion of Local Hospitals and University hospitals is not clear. Authors state that the inclusion of local hospitals is to reach patients who have not started renal replacement therapies. The inclusion of the university hospitals is to reach both pre-dialysis and dialysis patients. However, both local and university hospitals have large dialysis units.
8. It is also unclear how the authors define 'patients who are yet to start renal replacement therapy' and 'pre-dialysis patients'. In the introduction section, authors state 'local hospitals will target pre-dialysis patients'. There should a consistency between what is claimed in the introduction and methods sections.

	9. Though the authors describe that the educational intervention is carried out by 'allied-health transplantation professionals' in the abstract, this is not covered in the methods section.
--	---

REVIEWER	liise kayler University at Buffalo USA
REVIEW RETURNED	04-Sep-2018

GENERAL COMMENTS	this protocol describes a planned implementation study conducted by 8 Dutch hospitals where KTX candidates without living kidney donors will be enrolled to receive home-based living donor education. The outcome of cost effectiveness will be measured using a Markov Model and the outcome of adherence to the educational program will be measured using team meetings, coaching, and intermittent supervision of the educational sessions. The protocol is clearly written and the information derived from it is likely to be useful to the transplant community. Limitations were clearly discussed. However, Assessment of implementation of the study protocol is not robust. Perhaps recording the sessions and grading the teaching and advising that actually occurred would serve as a better indicator of intervention fidelity. other comments below. abstract; Costs data will be o... should be 'cost' data should be collected on the number of potential enrollees that do not consent to participate and the reason for nonparticipation. This is important for gleaning the proportion of patients interested in such programs to inform programs and stakeholders of the number of KTX candidates likely to be reached.
--

VERSION 1 – AUTHOR RESPONSE

Reaction to the reviewers' comments on the manuscript "Cost-effectiveness of Educational Interventions Targeting Living Kidney Donation" (bmjopen-2018-025684).

Review Comments:

Amy D. Waterman, PhD:

Primary concerns:

Primary concern No. 1 of Reviewer 1: The structure and organization of the paper causes confusion about what aspects of the methodology apply to this cost-effectiveness study, the implementation study in general, and the development of the education. Is this a protocol paper for the implementation study AND the cost-effectiveness analysis or only the cost-effectiveness analysis? If this paper is intended to serve as a protocol for the implementation study generally, more details are needed as listed here. If not, a concise summary of these items should be included.

a. Please provide references or information for what intervention is being implemented as an educational program. What was the educational program based on? Is it based upon the prior RCT

and Cross-Over studies mentioned? Is there a prior protocol for this education program implementation?

b. "The educators organize the intervention in such a way that they will do 'whatever it takes', in line with one of the basic principles of multisystem therapy (MST)." This statement does not have any references to a prior study nor a reference to a description of what "whatever it takes" principle is in MST.

Our reaction: We appreciate the remark about the structure and organization of the paper. We agree that the current manuscript can cause confusion because the manuscript both elaborates on the implementation study and the cost-effectiveness analysis. The manuscript is intended as a protocol for the cost-effectiveness analysis. The reason we discuss the implementation study in the method section, is because two previous interventions (reported in the RCT and Cross-Over studies) are merged into one in the implementation study. Therefore we gave a short overview of the most important features of the implementation study/combined intervention.

a. The implementation study is indeed based on the prior RCT and Cross-Over studies. These studies were based on the study of J. R. Rodrigue, as mentioned in the introduction section.

b. We agree that in the current form this statement is hollow. A reference to a study with a description of what "whatever it takes" means in MST is necessary.

Changes made: We discussed the parameters used in the cost-effectiveness analysis in more detail. See primary concern No. 2 for more detail. We have chosen to keep the information on the intervention as this is needed to understand the cost-effectiveness analysis.

Changes made: We emphasized in the method section that the implementation study was based on the previous studies.

In the method section:

"The intervention is based on the previous RCT and Cross-Over studies [3, 4, 11]."

Changes made: We added the following reference to the sentence:

7. Henggeler SW, Schoenwald SK, Borduin CM, Rowland MD, Cunningham PB: Multisystemic Therapy for Antisocial Behavior in Children and Adolescents. New York: The Guilford Press; 2009

Primary concern No. 2 of Reviewer 1: The details of the cost-effectiveness analyses provided are not sufficient to justify separate publication.

a. How is the data from the implementation study being used in the cost-effectiveness analysis? What data from the previous RCT will be used?

b. Why isn't the cost data from the literature included in this protocol? It is part of the input into the cost-effectiveness model.

c. How will the productivity costs be estimated?

d. How will the costs of the intervention and quality assurance be estimated?

e. How many simulations will be run for the Markov model? What are the assumptions that will be made? What are the estimated model parameters and distributions that will be used as input into the model? Which ones are based on the literature and which ones require data from the completed implementation study? These should be listed, perhaps in a table. What sensitivity analyses will be

conducted to evaluate robustness of results to incorrect assumptions? What software or programming language will be used to run these models?

f. The authors state “The outcome of the implementation will be compared with the baseline situation; the situation before this program was implemented.” The table of model inputs requested should clearly show what will be the same and will be different in these baseline and post-implementation models.

g. Here are two examples of cost-effectiveness analyses that provide much more detail on the model specifications:

<https://link.springer.com/article/10.1007%2Fs00198-014-2999-4>

<https://tobaccocontrol.bmj.com/content/23/3/223.long>

Our reaction: We appreciate Dr. Amy Waterman for the invitation to provide more details about the cost-effectiveness analysis. We agree that a protocol for a cost-effectiveness analysis should provide more detail.

a. In the implementation study we collected the health-related quality of life of patients participating in the educational program. In this way, we can attach a QALY to every health state that we use for the Markov Model. Furthermore, we collect protocol adherence scores given by the patients. We expect to have variation in the protocol adherence scores between the participating hospitals so we can make an estimation of the cost-effectiveness of the quality assurance system. Also, because 8 hospitals are participating in the implementation study, we can make a projection of the RRT-facilities needed in The Netherlands. Since we do not have a control group in the implementation study we are forced to use the effect size of the previously mentioned RCT.

b. The study on costs of dialysis and transplantation is currently in its final phase of publication according to the main researcher Dr. Ardine de Wit. Therefore, we cannot include the data in this protocol yet.

c. Productivity costs are estimated using the friction cost approach, as recommended by the Dutch Healthcare Institute (Zorginstituut Nederland). We record the work situation of patients in the implementation study. Furthermore, several reports are written in The Netherlands about the labor participation of patients with ESRD. We therefore can make an accurate estimation of the costs of the productivity losses of patients with ESRD in The Netherlands.

d. The implementation study is coordinated by a project group in the Erasmus Medical Center. The project group is also involved in this cost-effectiveness study. We therefore have the budget of the intervention per center and the budget for the quality assurance system in our possession.

e. The Markov model will run 10,000 simulations and will be built in Excel 2010. The main assumption that we make is that the effect size will be the same as found in the RCT. With sensitivity analyses we can find out what happens to the cost-effectiveness whether this assumption is either an over- or underestimation. We have to look in detail which distribution fits the best. For the costs a gamma distribution is most common in cost-effectiveness analyses and for the quality of life a beta distribution is most widely used. We do not yet have access to the database of Nefrovisie and we have to calculate the transition probabilities and do not know which distribution we have to use for the transition probabilities.

f. The only parameter that will change in the comparator are the transition probabilities, besides the additional costs of the educational programme. We expect more living donor kidney transplantations because of the programme and consequently a lower probability that patients switch between and to the dialysis modalities. We agree with the reviewer that the current manuscript lacks overview.

Therefore, we added a table (table 1) with an overview of the model parameters that we use and their sources.

g. We would like to thank the reviewer for the two examples given.

Changes made: We added a lot more detail of the cost-effectiveness analysis in the method section, see pages 9-11. We attempted to discuss every point raised by the reviewer in the method section. We also added a table with an overview of the different parameters.

Comments of Reviewer 1

Comment No. 1 of Reviewer 1: The paper is titled “Cost-effectiveness of Educational Interventions Targeting Living Kidney Donation”, implying that more than one intervention is being evaluated but this does not seem to be the case. Please clarify.

Our reaction: We agree that the current title is confusing.

Changes made: We have altered the title to:

“Cost-effectiveness of a home-based group educational programme on renal replacement therapies: A study protocol.”

Comment No. 2 of Reviewer 1: Is the projected sample size sufficient to address to the hypotheses?

Our reaction: For the first and third hypotheses we expect that the projected sample size will be more than sufficient, since the expected effect size of the programme is relatively large.

For the second hypothesis, we have made a power calculation. From the previous research, we know that the educational programme results in 4 times more donations when compared to standard care (reference date 13-11-2013). The proportions were then 0.088 versus 0.297 of the number of randomized patients. In the current implementation process there is no control group, all participants receive the educational intervention. However, differences between the teams can arise, for example, by differences in ‘treatment adherence’ to the protocol by the various teams or, for example, by complications in the communication between the team and the clinic, and differences in the standard care within which the programme is embedded. This implementation process will expose these differences. The differences are likely to be smaller than the 1:4 ratio as measured between the presence and the absence of the programme. That is why we need a larger number of patients. In this implementation process we are going to make a distinction between the home interventions with a good ‘treatment adherence’ and home interventions with a relatively ‘bad’ ‘treatment adherence’, separated by the median. If we rely on a ratio of 1 to 2 (in terms of proportions: 0.177 versus 0.297, or a duplication of the effect of the –old- control arm), then we need 195 patients for every group, or almost 400 in total (see summary below). This is based on an alfa of 0.05, a beta (power) of 0.80 and a two-sided test. The latter is a conservative assumption because we are actually just interested factors that influence effective implementation, thus a one-tailed test would be sufficient. We have chosen not to put this in the manuscript, because we think it would compromise the clearness of the manuscript.

Changes made: No changes made.

Comment No. 3 of Reviewer 1: Quality of life data from patients who are approached for the educational program—Is this using the EQ-5D-5L? When is the questionnaire being administered? The authors state that “we are currently in the process of collecting quality of life data from both pre-dialysis patients and dialysis patients”—is this referring to the implementation study or something separate?

Our reaction: Yes, we use the EQ-5D-5L to gather the quality of life data from patients who are approached for the educational programme. The questionnaire is administered prior to the intervention and 6, 12 and 24 months after the intervention. Thus, every patient is asked four times to fill out the questionnaire. This is part of the implementation study.

Changes made: We have emphasized this in the method section, under "Effects".

In method section:

We altered this sentence from:

"Moreover, we will also collect quality of life data of the patients who received the intervention"

To:

"We also collect quality of life data from patients after the intervention. The educators administer the EQ-5D-5L questionnaire to patients at 6, 12 and 24 months after the intervention by telephone."

Comment No. 4 of Reviewer 1: "Supervision and quality assessment were employed to map the generalizability of previous research." This meaning of this sentence is not clear.

Our reaction: We agree that this sentence is not clear.

Changes made: We have changed the sentence to:

"Supervision and quality assessment are being employed to maintain the quality of the intervention."

Comment No. 5 of Reviewer 1: What are the inclusion/exclusion criteria for the implementation study? Particularly for the pre-dialysis patients.

Our reaction: For the local hospitals that target pre-dialysis patients, the inclusion criteria are: ≥ 18 years of age, are eligible for transplantation, and primary RRT required within the coming 12 months. For the university hospitals that also target dialysis patients, the inclusion criteria are: ≥ 18 years of age, currently undergoing RRT or required within the coming 12 months and eligible for transplantation. We explicated this in the manuscript.

Changes made:

In the method section, on page 6:

"For these hospitals, the inclusion criteria are: ≥ 18 years of age, are eligible for transplantation, and primary RRT required within the coming 12 months."

"Eligible patients for the University hospitals are required to be ≥ 18 years, currently undergoing RRT or required within the coming 12 months and eligible for transplantation."

Comment No. 6 of Reviewer 1: The stated second objective is to investigate the relationship between the quality of the implementation of the program and outcome but no analysis methods are described to address this objective.

Our reaction: We agree that this should be mentioned in the method section how we aim to analyse this.

Changes made:

In the method section, on pages 11-12:

“Protocol adherence measures will be assessed by a third party by means of a telephone interview with the patients and the invitees. Patients and invitees are asked for their opinion and level of satisfaction regarding the way in which the intervention was delivered. Patients and invitees are asked to answer a 15-item scoring list. Items are rated on a Likert-scale (1 not at all – 5 very much). Only items that are rated with a 5, will be regarded as fully adherent. Items scored with a 1-4 will be regarded as non-adherent. The outcome of the protocol adherence will be compared with the gain in knowledge and communication skills. We will also look if there is a correlation between the protocol adherence and the amount of LDKTs.”

Comment No. 7 of Reviewer 1: The inclusion of Local Hospitals and University hospitals is not clear. Authors state that the inclusion of local hospitals is to reach patients who have not started renal replacement therapies. The inclusion of the university hospitals is to reach both pre-dialysis and dialysis patients. However, both local and university hospitals have large dialysis units.

Our reaction: The reason local hospitals only approach patients who have yet to start renal replacement therapies is because we applied the protocol of the Cross-Over study directly on the implementation study. The Cross-Over study was intended to reach the pre-dialysis patients to increase pre-emptive transplantations. The university hospitals are approaching both types of patients, as did the RCT-study. We agree that this should be more stated more clearly in the manuscript.

Changes made: We added an explanation of this distinction in the method section:

In the method section, on page:

“The distinction between the university hospitals and the regional hospitals is in line with the protocols of the abovementioned Cross-over study and the RCT.”

Comment No. 8 of Reviewer 1: It is also unclear how the authors define ‘patients who are yet to start renal replacement therapy’ and ‘pre-dialysis patients’. In the introduction section, authors state ‘local hospitals will target pre-dialysis patients’. There should a consistency between what is claimed in the introduction and methods sections.

Our reaction: We strongly agree that there should be consistency in the manuscript. We would like to thank the reviewer for pointing this out. We defined ‘patients who are yet to start renal replacement therapy’ and ‘pre-dialysis patients’ as the same group of patients. We have chosen to use the former wording in the manuscript, because pre-dialysis suggests that the first form of RRT will be dialysis.

Changes made: Throughout the manuscript we changed “pre-dialysis” to “patients who are yet to start renal replacement therapy”, see pages 5, 6 and 9.

Comment No. 9 of Reviewer 1: Though the authors describe that the educational intervention is carried out by ‘allied-health transplantation professionals’ in the abstract, this is not covered in the methods section.

Our reaction: We agree that this is an inconsistency. We therefore mention that the intervention is carried out by allied-health transplantation professionals and psychologists in both the abstract and the method section.

Changes made: We attempted to make the manuscript more consistent and changes two sentences in the abstract and method section.

In the abstract:

“This is carried out by allied-health transplantation professionals and psychologists across 8 hospitals in The Netherlands.”

In the method section on page 7:

“The intervention is carried out by allied-health transplantation professionals and psychologists.”

Liise Kayler, PhD:

This protocol describes a planned implementation study conducted by 8 Dutch hospitals where KTX candidates without living kidney donors will be enrolled to receive home-based living donor education. The outcome of cost effectiveness will be measured using a Markov Model and the outcome of adherence to the educational program will be measured using team meetings, coaching, and intermittent supervision of the educational sessions. The protocol is clearly written and the information derived from it is likely to be useful to the transplant community. Limitations were clearly discussed. However, Assessment of implementation of the study protocol is not robust. Perhaps recording the sessions and grading the teaching and advising that actually occurred would serve as a better indicator of intervention fidelity.

Our reaction: We are glad that the reviewer appreciates our work. We agree that there are better indicators of intervention fidelity. We do, however, let the educators grade the teaching and advising that occurs every 6 weeks. We have a 10-item scoring list that the educators fill out regarding the content and interpersonal delivery of the supervisor in these meetings. The educators have also room to express their concerns or other remarks in the questionnaire. We will take into account these scores when we assess the quality assurance system. Currently we are not recording the supervision sessions, but perhaps we should consider doing so in the future. We would like to thank the reviewer for this suggestion.

Changes made: We have added the following sentence:

In the method section on page 12:

“After these meetings the supervisor is graded by the educators through a 10-item questionnaire regarding the content of the teaching and the interpersonal delivery of the supervisor.”

Comment No. 2 of Reviewer 2: abstract; Costs data will be o... should be 'cost'.

Our reaction: We have corrected this error.

Changes made: In the abstract under section methods and design, we have changed the word “costs” to “cost”.

Comment No. 2 of Reviewer 2: data should be collected on the number of potential enrollees that do not consent to participate and the reason for nonparticipation. This is important for gleaning the proportion of patients interested in such programs to inform programs and stakeholders of the number of KTX candidates likely to be reached.

Our reaction: We strongly agree with the reviewer. Every educator is asked to record every patient they approach for the study. If a patient does not consent to participate in the programme, the patient is asked why she/he does not want to participate. Furthermore, we record the day of birth and gender of these patients. This information could also be of value for policy-makers and health insurers.

Changes made: We decided to mention this in the article.

In the method section on page 7:

“The amount of patients that do not consent to participate in the study is recorded, as well as the reason for nonparticipation.”